# The Responses of Physiological Characteristics and Flowering Related Gene to the Different Water Stress Levels of Red-Flesh Pummelo Cultivars (*Citrus grandis* (L.) Osbeck) Own-Rooted by Air Layering Propagation under Two Growing Conditions

Prawit Thammatha [1], Chanon Lapjit [1,2], Tanyarat Tarinta [1,2], Sungcom Techawongstien [1,3] and Suchila Techawongstien [1,2,*]

[1] Horticulture Section, Faculty of Agriculture, Khon Kaen University, Khon Kaen 40002, Thailand; prawitt@kkumail.com (P.T.); lchano@kku.ac.th (C.L.); tanyata@kku.ac.th (T.T.); suntec@kku.ac.th (S.T.)
[2] Plant Breeding Research Center for Sustainable Agriculture, Faculty of Agriculture, Khon Kaen University, Khon Kaen 40002, Thailand
[3] Research Group for Fruit Crops in the Northeast, Faculty of Agriculture, Khon Kaen University, Khon Kaen 40002, Thailand
* Correspondence: suctec@kku.ac.th

**Abstract:** One of the major problems in the fruit production of citrus, including pummelo (*Citrus grandis*) is controlling flowering induction. Water stress is known to be related to flowering induction via physiological responses related to the flowering gene. However, reports on the mechanisms underlying floral induction by water stress in pummelo are limited. Thus, this study aimed to determine the physiological characteristics and the expression of genes related to flowering induction, *CiFT* (*Citrus Flowering locus T*), in pummelo at different levels of water stress. Experiments were conducted under two growing conditions: field and container conditions, each using a $2 \times 5$ factorial experiment in a randomized complete block. Factor A consisted of two red-flesh pummelo cultivars while factor B consisted of five levels of water stress based on the leaf rolling index. Among the seven characteristics studied, only the data of total nitrogen, *CiFT*, and flower number were combined for analysis due to their results in a homogeneity test. Although a consistent tendency was not observed for the interaction among environments, genotypes, and water stress levels of all characteristics, 'KKU-105' grew more flowers under higher water stress conditions (225 flowers). This result may imply that decreases in total nitrogen (1.48%), stomatal conductance (50.53 $m^{-2}s^{-1}$), chlorophyll fluorescence (0.30 Fv/Fm), and upregulation of *CiFT* mRNA level (13.95) may induce flowering in the pummelo cultivar 'KKU-105'.

**Keywords:** container growing condition; environment; water deficit; drought stress; floral induction; mRNA; key signal; stomatal conductance; chlorophyll fluorescence

## 1. Introduction

In Thailand, Pummelos (*Citrus grandis* (L.) Osbeck) are divided into two kinds based on the color of their flesh: white-flesh pummelos and red-flesh pummelos [1]. Currently, the red-flesh pummelo is popular among consumers due to its richness in lycopene, anthocyanin, and beta carotene [2–4]. The red-flesh pummelo cultivars ('KKU-105', 'KKU-101', and 'Manee Esan') are some of Thailand's newest and most promising pummelo cultivars, and can be found in Baan Bung 14, Tambon Nonthong, Amphoe Kasetsomboon, Chaiyaphum province in the northeast. However, red-flesh pummelos grown in tropical and sub-tropical areas are frequently susceptible to climate change, which causes their flowers to fall off owing to high relative humidity and rainfall [5]. Moreover, stress related conditions can directly and indirectly decrease the expression of genes which are responsible for floral induction or flowering, which is one of the most crucial events in a plant's lifecycle [6,7].

Several genes have been reported to be linked to flowering, namely *FLOWERING LOCUS T (FT)* gene, which is one of the most studied genes and is typically maximally expressed at the onset of floral induction [8]. *FT* homologs are also the key regulators of flower development in plants belonging to the Citrus genus [9–14]. A decrease in the expression of these genes related to flowering would hinder flower development, thus negatively affecting the yield for pummelo trees.

In general, flowering is influenced by environmental factors such as temperature, light, moisture, nutrients, and drought stress, all of which are linked to changes in crop plant gene expression [15]. Plants could avoid water stress by different morphological, biochemical, and molecular mechanisms [16]. Stress in particular plays a key role in con-trolling plant flowering [17,18]. While citrus trees are subjected to floral inductive temperatures during the winter in the subtropics, tropical areas do not experience significant seasonal temperature variations. Instead, floral induction is linked to water stress during the dry season instead of changes in temperature. In fact, among the environmental stresses, water stress is the main factor that affects agricultural production, particularly in irrigated land. Water stress induces both flowering and the upregulation of the homolog of *FT* gene in sweet orange (*Citrus sinensis*) [13,19]. Moreover, the *Citrus FLOWERING LO-CUS T (CiFT)* gene expression level was found to be correlated with floral induction under the water stress condition [18,20]. Besides, severe water stress produced maximum flowering capac-ity compared to mild water stress in the Tahiti lime (*Citrus latifolia*) [21]. The water stress or water deficit stressful conditions may degrade necessary physiological processes such as stomatal conductance, photosynthesis, and nitrogen uptake [22–29]. A major reduction in chlorophyll fluorescence was also caused by water deficiency, suggest-ing photosynthesis inhibition [30]. In addition, these on the whole enhanced carbohydrate accumulation in the bud as well as promoted flowering. Flowering is positively correlated with carbohydrate levels in the bud and phloem [30,31].

Since field conditions provide an unfavorable environment which strongly affects floral induction, it is necessary to cultivate crops under controlled container conditions so as to avoid any biotic (pest and disease) and abiotic (rainfall and temperature) stresses. However, the plant is generally grown under a limited container, which decreases shoot growth and photosynthesis because of the similarity of physiological mechanisms to those of water stress [32,33]. Nevertheless, information on physiological responses associated with the flowering-related gene under limited growing conditions in citrus is still scarce. An understanding of the association between water stress and flowering is essential for enhancing flowering induction in citrus. In this study, red-flesh pummelo were subjected to drought stress in a growth container (potted) and field condition. The goal of determine the mechanism of the effect of different water stress levels on the flowering gene based on the physiological response of pummelo. Therefore, the aim of this study is to determine the responses of physiological characteristics to expression *CiFT* mRNA level and flowering in two red-flesh pummelo cultivars at different levels of water stress under two settings (container-grown (potted) and field-grown).

## 2. Materials and Methods

### 2.1. Plant Materials and Treatments

Two red-flesh pummelo cultivars, i.e., 'KKU-105' and 'Manee Esan' own-rooted by air layering propagation (both 5 years old) were used. They were grown under two different conditions, i.e., container and field conditions. The container-grown (plastic pots) under plant house, the pummelo plants were transplanted into plastic pots during the months of March–April 2014. Plastic pots (100 L 0.53 m $\times$ 0.72 m; W $\times$ H) were filled with a mixed media consisting of rice husk, rice husk charcoal, peat moss, and cow manure in a ratio of 2:1:1:0.5 *(v/v)*, respectively. Field-grown pummelo plants were transplanted into a field with a spacing of 6 $\times$ 6 m during the months of March–April 2014.

Under both settings, pummelo plants were watered daily at field capacity throughout the experiment, except during the treatment period. which lasted for 2 months. Every two

weeks throughout the experiment, 1.0 kg of organic fertilizer (cow manure) and 0.5 kg of inorganic fertilizers (12-24-12/N-P-K) were applied to each plant, except during the treatment period. The study was carried out with field- and container-grown (plastic pots) plants at the experiment farm of Khon Kaen University, Thailand (latitude 16°21.145′ N, longitude 101°48.916′ E) during the months of October to November 2018.

The experiment was conducted using a 2 × 5 factorial in a randomized complete block design with three replications under each growing condition (Figure 1). Factor A consisted of two red-flesh pummelo cultivars ('KKU-105' and 'Manee Esan'), three plants of each cultivar were used in each replication. Factor B consisted of five levels of water stress based on a leaf rolling index, i.e., unrolled leaves in the control group (WS1), folded deep-V-shaped leaves (WS2), fully-cupped U-shaped leaves (WS3), margin-touching O-shaped leaves (WS4), and tightly-rolled leaves (WS5) [34–36]. Measurements were recorded on the last day of the water stress period and after re-watering at 10:00 and 14:00 h

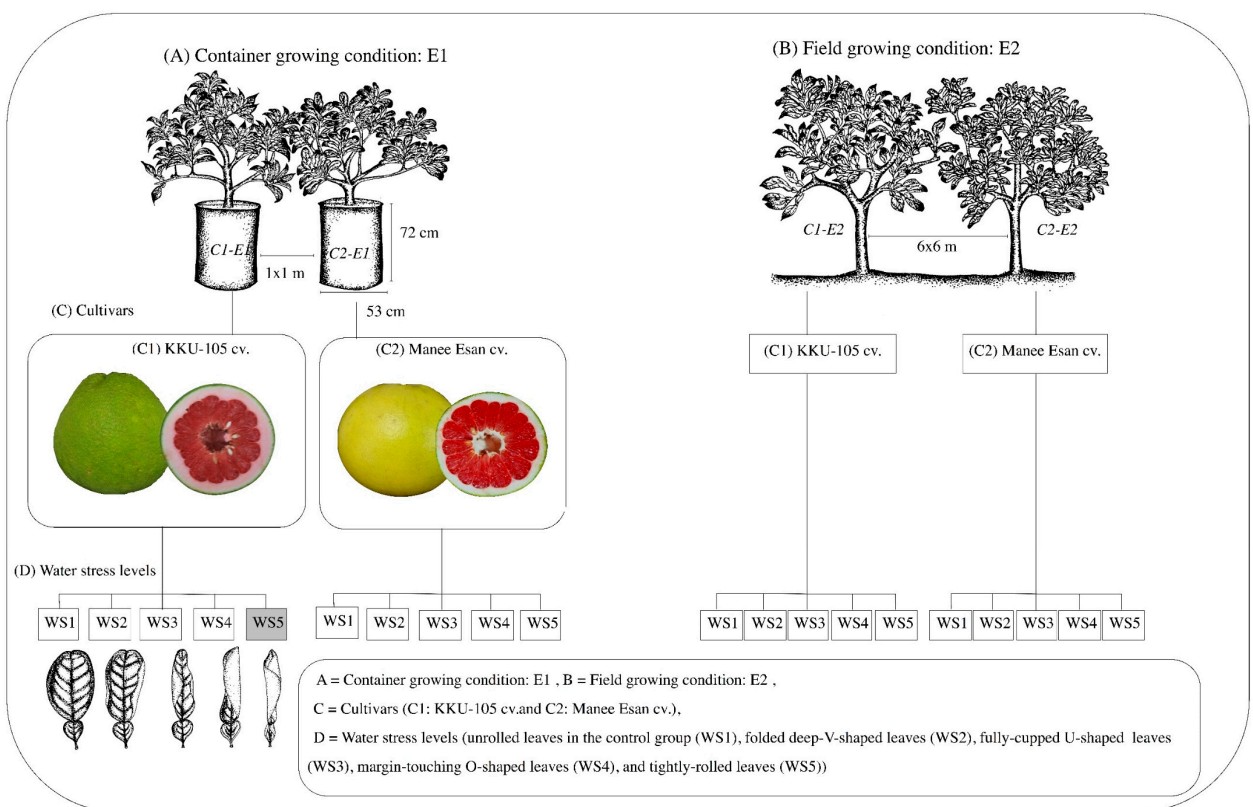

**Figure 1.** The schematic procedure for flowering induction of two different red-flesh pummel cultivars grown at five different water stress levels in two environments (container and field conditions).

### 2.2. Physiological Characteristics Measurement

Stomatal conductance (SC) and chlorophyll fluorescence (CF) were measured at the middle of the canopy from three fully exposed mature leaves per plant. Stomatal conductance (mmol m$^{-2}$ s$^{-1}$), measured by a porometer, is the rate of $CO_2$ entering, or water vapor exiting through stomata. A porometer (Model AP4, Delta-T Devices Ltd., Cambridge, UK) was used to measure stomatal resistance (s/cm). CF data were collected using a Handy PEA chlorophyll fluorometer (Hansatech, Kings Lynn, UK). Air temperatures and relative humidity were recorded daily using a data logger (HOBO U12 Data Loggers, Onset Computer Corporation, Bourne, MA, USA). The mature leaves at the upper portion of the plant were sampled, then immediately frozen in liquid nitrogen and stored at −80 °C until use. The 100 g of fresh leaves were dried in freeze dryers at −53 °C for 48 h (Scanvac cool

safe 55-9 Model). The dried leaves were ground in a blender and stored at $-20\,^{\circ}$C until further analysis.

Soluble carbohydrates were extracted from 250 mg of ground material by diluted sulfuric acid (0.2 N $H_2SO_4$) as described [37]. The extracted solutions were measured using a spectrophotometer for TNC concentration by applying the phenol sulfuric acid using the Nelson-Somogyi method citation.

Soluble extract of 300 mg of the ground material in 98% $H_2SO_4$ were obtained by the method described [38]. TN concentration of the extract was determined based on the UV oxidation method proposed [39].

### 2.3. Genetic Characteristics Measurement

#### 2.3.1. RNA Extraction and cDNA Synthesis

Total RNAs were extracted from leaves using PureLink® Plant RNA Reagent (Thermo Fisher Scientific, Waltham, MA, USA) according to the manufacturer's instructions. cDNA synthesis reactions were performed using ReverTra Ace®qPCR RT Master Mix with gDNA Remover (TOYOBO, Osaka, Japan). The cDNA product was stored at $-20\,^{\circ}$C.

#### 2.3.2. Gene Expression Analysis

Gene expression levels were measured with reverse-transcription-quantitative real-time PCR (RT-qPCR) using the StepOnePlus Real-Time PCR system (Thermo Fisher Scientific, Waltham, MA, USA) by comparative CT method. Gene amplification was performed using 10 ng of working cDNA solution. Each reaction mixture comprised 2 uL of cDNA, 10 uL of THUNDERBIRD® SYBR® qPCR Mix (TOYOBO, Osaka, Japan), 6.8 uL of RNase-free water, 0.6 uL of forward, and 0.6 uL reverse primers. The primers used for *CiFT* amplification was designed as (F:GGGAGGCAGACTGTTTATGC and R:CGGAGGTCCCAGATTGTAAA) [12]. The endogenous control reference genes, FBOX and SAND were used taken [40] with a final primer concentration that was later diluted four-fold for the subsequent analysis. A negative control containing all RT-qPCR reaction elements and water instead of cDNA were used for every 96-well reaction plate. The parameters for reactions were set as follows: one cycle of pre-denaturation at 95 °C for 20 s; 40 cycles of denaturation at 95 °C for 20 s, annealing at 60 °C for 20 s, and extension at 72 °C for 40 s.

### 2.4. Statistical Analysis

Data of each parameter from each growing condition was statistically analyzed by using a 2 × 5 factorial in randomized complete block design. The variances were tested for homogeneity, then data were combined for analysis [41]. Treatment means were separated by the least significant difference (LSD) test at a 5% probability level ($p < 0.05$).

## 3. Results

During the experiment (October to November 2018), the monthly maximum air temperature, minimum air temperature, average relative humidity, and total rainfall (for both container- and field-grown plants) were almost similar, i.e., 36.1 °C, 16.7 °C, 66.2%, and 3.8 mm, respectively (Figure 2). In terms of homogeneity, the ratios between large EMSs (error mean square) and small EMSs that were found to be higher than three were the SC, CF, TNC, and C:N ratios. On the other hand, the ratios of TN, *CiFT* mRNA level, and the number of flowers were found to be less than three (data not shown). The genotype-by-water stress (G × S) interactions were significantly different among the four characteristics studied (Table 1). The mean values of SC value and CF value in container-grown plants (131.87 mmol $m^{-2}s^{-1}$ and 0.61 Fv/Fm, respectively) were lower than those in field-grown plants (136.89 mmol $m^{-2}s^{-1}$ and 0.71 Fv/Fm, respectively). Contrastingly, the mean values of the TNC and C:N ratio value in container-grown plants (151.87 mg/g DW and 7.30% DW, respectively) were higher than those in field-grown plants (144.56 mg/g DW and 6.10% DW, respectively).

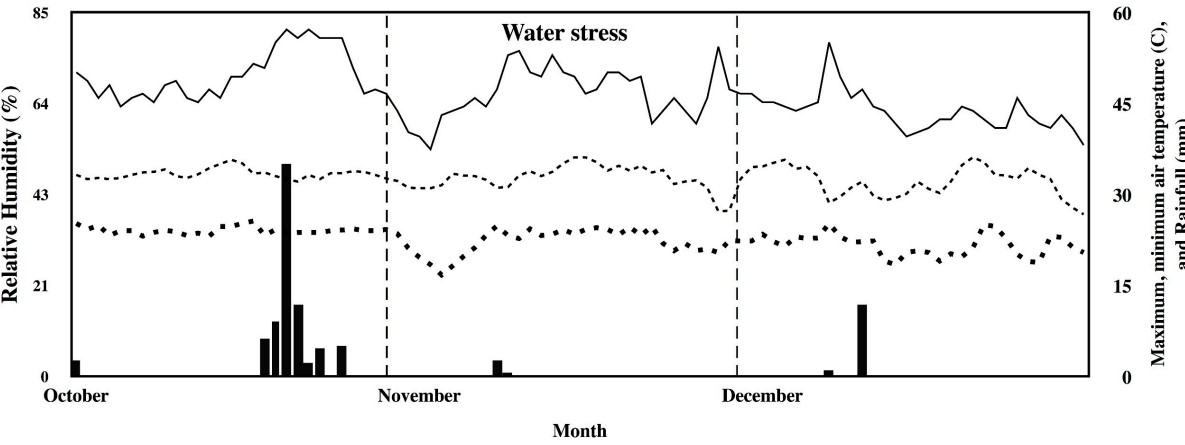

**Figure 2.** Monthly mean air temperature, relative humidity, and rainfall in plant house (container-grown) and at the experiment farm (field-grown), Khon Kaen University during a season (October to December 2018).

**Table 1.** Stomatal conductance, chlorophyll fluorescence, total non-structural carbohydrates, and total non-structural carbohydrates per total nitrogen (C: N ratio) of two red-flesh pummelo cultivars grown under five different water stress levels and two settings (container-grown: E1 and field-grown: E2).

| Treatments | | Stomatal Conductance (mmol m$^{-2}$s$^{-1}$) | | Chlorophyll Fluorescence (Fv/Fm) | | Total Non-Structural Carbohydrates (mg/g DW) | | C:N Ratio (% DW) | |
|---|---|---|---|---|---|---|---|---|---|
| | | E1 | E2 | E1 | E2 | E1 | E2 | E1 | E2 |
| Genotype (G) | | | | | | | | | |
| KKU-105 | | 129.70 | 135.15 | 0.57 b | 0.67 b | 154.60 a | 148.28 a | 7.08 a | 6.69 a |
| Manee Esan | | 134.04 | 139.30 | 0.65 a | 0.74 a | 149.15 b | 140.84 b | 6.80 b | 5.53 b |
| Water stress level (WS) | | | | | | | | | |
| WS1 | | 352.11 a$^y$ | 352.95 a | 0.85 a | 0.86 a | 129.20 e | 129.30 e | 4.66 e | 4.67 e |
| WS2 | | 111.76 b | 124.85 b | 0.76 b | 0.81 b | 139.17 d | 137.14 d | 5.35 d | 4.95 d |
| WS3 | | 86.40 c | 89.68 c | 0.63 c | 0.71 c | 153.71 c | 142.98 c | 6.99 c | 5.66 c |
| WS4 | | 58.97 d | 64.49 d | 0.47 d | 0.60 d | 163.52 b | 155.76 b | 8.81 b | 7.49 b |
| WS5 | | 50.13 e | 54.15 d | 0.35 e | 0.55 e | 173.79 a | 157.63 a | 10.75 a | 7.76 a |
| Interaction (G × WS) | | | | | | | | | |
| KKU-105 | WS1 | 354.11 b | 358.97 a | 0.85 a | 0.88 a | 129.73 f | 129.78 f | 4.75 g | 4.71 g |
| | WS2 | 106.04 d | 115.00 c | 0.73 c | 0.80 b | 140.67 e | 136.16 a | 5.45 f | 4.94 f |
| | WS3 | 86.04 e | 87.00 e | 0.61 e | 0.71 c | 155.86 cd | 143.84 d | 7.20 d | 5.68 e |
| | WS4 | 60.97 f | 62.80 f | 0.36 h | 0.52 e | 168.55 b | 164.32 b | 9.61 b | 8.89 b |
| | WS5 | 50.35 g | 51.97 g | 0.30 i | 0.45 f | 178.18 a | 167.32 a | 12.00 a | 9.22 a |
| Manee Esan | WS1 | 359.10 a | 346.93 a | 0.86 a | 0.83 b | 128.66 f | 128.82 f | 4.56 g | 4.63 g |
| | WS2 | 17.47 c | 134.70 c | 0.79 b | 0.82 b | 137.67 e | 138.12 e | 5.25 f | 4.97 f |
| | WS3 | 86.76 e | 92.37 d | 0.64 d | 0.72 c | 151.56 d | 142.12 d | 6.78 e | 5.65 e |
| | WS4 | 56.97 g | 62.80 f | 0.59 f | 0.69 cd | 158.49 c | 147.19 c | 8.01 c | 6.10 d |
| | WS5 | 49.90 g | 56.34 g | 0.40 g | 0.65 d | 169.39 b | 147.95 c | 9.49 b | 6.30 c |
| | Mean | 131.87 B$^x$ | 136.89 A | 0.61 B | 0.71 A | 151.87 A | 144.56 B | 7.30 A | 6.11 B |
| Genotype (G) | | ns | ns | ** | ** | ** | ** | ** | ** |
| Water stress level (WS) | | ** | ** | ** | ** | ** | ** | ** | ** |
| Interaction (G × WS) | | ** | ** | ** | ** | ** | ** | ** | ** |
| CV (%) | | 3.59 | 7.52 | 1.74 | 3.08 | 1.40 | 0.77 | 2.08 | 1.33 |

ns, ** not significant and significant at 0.01 probability levels, respectively; the number within the parentheses is relative percentage of sum squares to total sum of squares. x: Means within a row followed by the same capital letter are not significantly different between growing condition at $p < 0.05$ by LSD. y: Means within a column followed by the same capital letter are not significantly different at $p < 0.05$ by LS.

For both cultivars and under both settings, it was found that the SC and CF values of interaction (G × WS) were low at severe water stress levels (WS5). The SC values in both cultivars, i.e., 'KKU-105' and 'Manee Esan' in container-grown plants (50.35 and 49.90 m$^{-2}$s$^{-1}$, respectively), and those cultivars in field-grown plants (51.97 and 56.34 mmol m$^{-2}$s$^{-1}$, respectively) were low at WS5. On the contrary, the CF values in the container-grown 'KKU-105' (0.30 Fv/Fm) were obviously lower than those of 'Manee Esan' (0.40 Fv/Fm) at WS5, as well as in field-grown plants (0.45 and 0.65 Fv/Fm, respectively). In terms of the responses of both cultivars under each growing condition, the TNC and C:N ratio values increased during severe water stress levels. Under both settings, the TNC and C:N ratio values of 'KKU-105' were higher than those of 'Manee Esan'. Furthermore, of the container-grown plants, 'KKU-105' at WS5 gave the highest TNC and C:N ratio value accumulation (178.18 mg/g DW and 12.00% DW, respectively). Similarly, the field-grown 'KKU-105' at WS5 gave the highest accumulated TNC and C:N ratio values (167.32 mg/g DW and 9.22% DW, respectively). However, the control treatment of both container- and field-grown cultivars gave the lowest accumulated TNC (129.20 and 129.30 mg/g DW, respectively) and C:N ratio (4.66 and 4.67% DW, respectively) values compared with the stress treatments.

Three characteristics, i.e., TN, *CiFT* mRNA levels, and the number of flowers with the ratio between large EMS and small EMS that were less than three were combined for a comparative analysis between two settings (Table 2). It was found that genotype, water stress levels, environment-by-water stress levels of interaction, and genotype-by-water stress levels of interaction were all highly significant ($p \leq 0.01$). Genotype-by-environment interaction levels were significantly different for the TN. In addition, the interactions among environment, genotype, and water stress levels were highly significant for all characteristics studied. Additionally, the effects of water stress levels accounted for a large portion (major effect) of the total variations for TN (54.70%), *CiFT* mRNA level (83.96%), and the number of flowers (89.88%).

**Table 2.** Mean squares for total nitrogen, CiFT mRNA level, and flower number of two red-flesh pummelo cultivars grown under five different water stress levels and two settings.

| Source of Variation | d.f. | Total Nitrogen (%) | | *CiFT* Gene | | Flower Number | |
|---|---|---|---|---|---|---|---|
| Environment (E) | 1 | 84.49 × 10$^{-2}$ ** | (23.60) | 7.49 ** | (2.62) | 3010.40 * | (3.38) |
| Reps. within E | 4 | 0.03 × 10$^{-2}$ | (0.01) | 11.90 | (0.04) | 191.20 | (0.21) |
| Genotype (G) | 1 | 51.15 × 10$^{-2}$ ** | (14.29) | 18.70 ** | (6.55) | 2318.80 ** | (2.60) |
| G × E | 4 | 2.40 × 10$^{-2}$ * | (0.67) | 0.01 $^{ns}$ | (0.01) | 79.70 $^{ns}$ | (0.80) |
| Error (a) | 4 | 0.24 × 10$^{-2}$ | (0.06) | 0.05 | (0.02) | 127.90 | (0.14) |
| Water stress level (WS) | 4 | 195.00 × 10$^{-2}$ ** | (54.70) | 239.48 ** | (83.96) | 80017.50 ** | (89.88) |
| E × WS | 4 | 8.69 × 10$^{-2}$ ** | (2.42) | 9.28 ** | (3.25) | 1040.90 ** | (1.16) |
| G × WS | 4 | 12.11 × 10$^{-2}$ ** | (3.38) | 9.09 ** | (3.18) | 2015.50 ** | (2.26) |
| E × G × WS | 4 | 2.78 × 10$^{-2}$ ** | (0.77) | 0.80 ** | (0.28) | 144.10 * | (0.16) |
| Error (b) | 32 | 0.19 × 10$^{-2}$ | (0.05) | 0.16 | (0.05) | 75.2 | (0.08) |
| CV a (%) | | 2.12 | | 3.78 | | 12.45 | |
| CV b (%) | | 1.85 | | 6.55 | | 9.55 | |

ns, *, ** not significant and significant at 0.05 and 0.01 probability levels, respectively; the number within the parentheses is relative percentage of sum squares to total sum of squares.

Under both settings, the TN values of both cultivars decreased while the *CiFT* mRNA level and the number of flowers increased at severe water stress levels (WS4-5) (Figure 3). Under container-grown, TN value of 'KKU-105' (1.48%) was lowest at WS5, followed WS4 (1.75%) (Figure 3A). The total nitrogen values of 'KKU-105', at severe water stress levels, were lower than those of 'Manee Esan' in both settings. In addition, the control treatment of 'KKU-105' and 'Manee Esan' gave the lowest accumulated TN value in the container-grown (2.72 and 2.81%, respectively) and field-grown (2.81 and 2.78%, respectively) compared with the stress treatments. In particular, the *CiFT* mRNA levels of 'KKU-105' at severe

water stress levels were higher than those of 'Manee Esan' (Figure 3B). However, under container-grown, 'KKU-105' at WS5 gave the highest *CiFT* mRNA level (13.95) and the highest number of flowers (225 flowers), followed by WS4 (197 flowers) (Figure 3C). In addition, the number of flowers in 'KKU-105' were higher than those of 'Manee Esan' at WS5 and WS4. However, the control treatment of 'KKU-105' and 'Manee Esan' gave the lowest accumulated *CiFT* mRNA levels when container-grown (0.24 and 0.18) and field-grown (0.19 and 0.10) compared with the stress treatments.

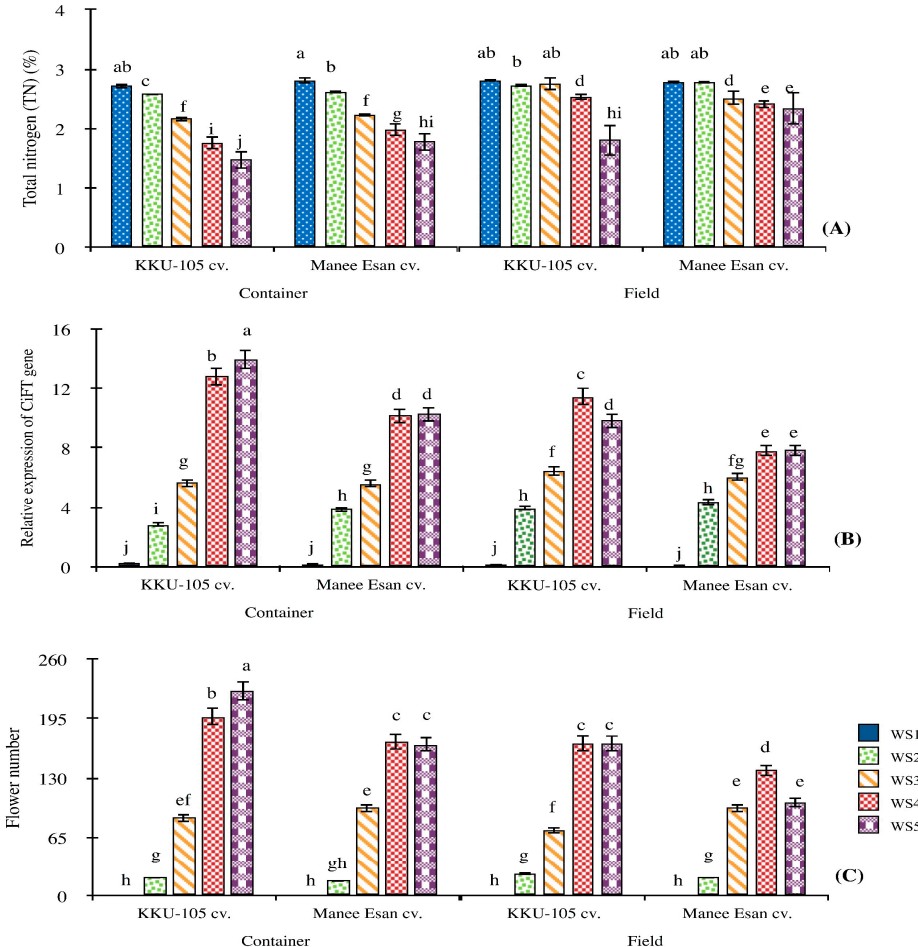

**Figure 3.** Total nitrogen (**A**), CiFT mRNA level (**B**), and flower number (**C**) of two red-flesh pummelo cultivars grown under five different water stress levels in two settings.

With regard to linear correlation analysis, significant differences at $p \leq 0.01$ were observed for all characteristics (Table 3). The correlation coefficients among water stress levels were high and positively correlated with TNC, C:N ratio, *CiFT* mRNA level, and the number of flowers (r = 0.816, 0.859, 0.948, and 0.965, respectively). However, the SC, CF, and TN were also high but negatively correlated with water stress levels (r = −0.842, −0.979, and −0.847, respectively). The number of flowers was high and positively correlated with water stress levels, TNC, C:N ratio, and *CiFT* mRNA level (r = 0.965, 0.719, 0.802, and 0.934, respectively). Nevertheless, the number of flowers was also high but negatively correlated with SC, CF, and TN (r = −0.753, −0.949, and −0.775, respectively).

**Table 3.** Linear correlations between the content of water stress levels, biochemical, flowering and expression analysis of flowering related genes.

| Characteristics | WS | SC | CF | TNC | TN | C:N ratio | *CiFT* mRNA |
|---|---|---|---|---|---|---|---|
| SC | −0.842 ** | | | | | | |
| CF | −0.979 ** | 0.845 ** | | | | | |
| TNC | 0.816 ** | −0.944 ** | −0.826 ** | | | | |
| TN | −0.847 ** | 0.959 ** | 0.860 ** | −0.910 ** | | | |
| C:N ratio | 0.859 ** | −0.941 ** | −0.873 ** | 0.933 ** | −0.972 ** | | |
| *CiFT* mRNA | 0.948 ** | −0.854 ** | −0.942 ** | 0.827 ** | −0.869 ** | 0.871 ** | |
| Flower | 0.965 ** | −0.753 ** | −0.949 ** | 0.719 ** | −0.775 ** | 0.802 ** | 0.934 ** |

** = Significant at 0.01 probability. Stomatal conductance (SC), chlorophyll fluorescence (CF), total non-structural carbohydrates (TNC), total nitrogen (TN), TNC:TN ratio (C:N ratio), and *CiFT* mRNA level.

## 4. Discussion

This study revealed that the number of flowers and *CiFT* mRNA level for floral induction in 'KKU-105' cultivated container-grown were obviously higher at severe water stress levels when compared to those cultivated at low water stress levels. This might have occurred as a result of the three subsequent responses. These are the decrease of TN, SC, and CF; the increase of TNC and C:N ratio; and the increase of *CiFT* mRNA level. These phenomena resulted in the highest number of flowers in this cultivar under the particular stress growing condition. Nevertheless, the number of flower and *CiFT* mRNA level in 'Manee Esan' were low at any water stress level when cultivated under both settings, which imply that response to water stress is genotype-dependent in red-flesh pummelo [41].

For the first responses, TN, SC, and CF decreased under severe water stress levels. These phenomena might be explained by the physiological metabolism in the plant. During water uptake, essential elements are generally transported in water flux through the whole plant caused by transpiration [42]. Nitrogen is recognized as one of the most important nutrients for plant growth and development [43,44]. However, the uptake of con-current nitrogen and water under stress conditions is somehow decreased [45,46]. Under such condition, low nitrogen retards the vegetative growth and promotes reproductive growth [47]. In addition, the phenomenon of stomata closure or low SC normally occurs in order to reduce transpiration and can be used as an indicator of plant water status [48,49]. Meanwhile, stomatal closure decreases photosynthesis under water stress [28,29,50]. Furthermore, CF has been generally used as an indicator of photochemical efficiency [51]. Thus, the low CF influences photoinhibition, which implies damage of photosystem II (PSII) [52,53]. Therefore, the drastic decrease of TN, SC, and CF in 'KKU-105' under severe stress condition compared to the other treatments was a result of the low nitrogen uptake and photosynthesis reduction during stomatal closure [22,54].

Regarding the second responses, the increase of TNC and C:N ratio started from the influences of TNC biosynthesis on the upregulation of the C:N ratio, which affected the greater relative expression of *CiFT* mRNA level under severe stress conditions. It has been found in previous studies that growing plants under severe stress conditions, such as growing them in containers with limited water application, aggressively suppresses their root and shoot growth [55,56]. In addition, the severe stress conditions have been found to reduce photosynthetic activity and transportation of its products [57] and has also been found to cause high accumulated transient storage of sucrose or TNC in the vacuole [58]. The high TNC is generally recognized as a critical criterion for initiating or promoting flowering [59,60]. The greater decrease of TN and increases of TNC in 'KKU-105' under severe stress compared to mild stress conditions may indicate lower nitrogen uptake up-on severe stress, which further leads to an accumulation of carbohydrates in the leaves of olive [61]. After that, the speeding up of carbohydrates accumulation decreases photosynthetic proteins (including rubisco) leading to a further decrease in leaf nitrogen and photosynthesis [62]. Furthermore, these mechanisms increased the C:N ratio in the leaves [63].

For the third response, high values of *CiFT* mRNA level generally induced the high number of flowers in citrus [14]. From our results, it was interesting to note that the high numbers of flower and *CiFT* mRNA level were clearly observed only in 'KKU-105' at severe water stress in container-grown. It has been recently reported that the *CiFT* gene plays an important role in the floral induction of citrus [14,64]. The *CiFT* gene expression and transcription are precisely regulated and exist in the mature leaves [65–67]. In addition, it has been found that this pathway is enhanced through the degradation of TNC into sucrose molecules in leaves after re-watering [68–71]. In particular, the high sucrose normally has an important function for increasing *FT* expression [72,73]. The *FT* gene is normally expressed into two forms, i.e., *FT* mRNA and *FT* protein [74]. Then, *FT* mRNA and *FT* proteins transport independently or combine to form what is called an *FT* RNA-protein complex before transported to the apical meristem [11,75,76]. The formation of floral induction will finally be induced by the increase in *FT* mRNA [8,77].

Nevertheless, the number of flower and *CiFT* mRNA level in 'Manee Esan' were low at both settings and any stress level. This was perhaps a result from the fact that such stress treatments were below a critical threshold for this cultivar. On the other hand, the mild stress levels including the treatments in field-grown did not reach the critical thresh-old of both pummelo cultivars.

## 5. Conclusions

The dramatic decrease of TN (1.48%), SC (50.53 $m^{-2}s^{-1}$), and CF (0.30 Fv/Fm), together with the increase of TNC (167.32 mg/g DW) and C:N ratio (9.22% DW) were observed in 'KKU-105' under severe stress condition. The increase of TNC and C:N ratio may upregulate the *CiFT* mRNA level (13.95). The association between the expression of *CiFT* mRNA level and the levels of water stress was observed in 'KKU-105'. These indicate that decreases in total nitrogen, stomatal conductance, chlorophyll fluorescence, and upregulation of *CiFT* expression may induce flowering induction in pummelo 'KKU-105'. That the highest number of flowers were obtained from 'KKU-105' (225 flowers) at severe water stress level in container-grown conditions implies that the production of red-flesh pummelo cultivars can be managed for quantity over the required period.

**Author Contributions:** Author contribution conceptualization and supervision: S.T. (Sungcom Techawongstien), S.T. (Suchila Techawongstien), P.T.; experimental design, instrument operation, statistical analysis: S.T. (Sungcom Techawongstien), S.T. (Suchila Techawongstien), T.T., C.L. and P.T.; assisting in sample preparation and RNA extraction: C.L.; all experiments: P.T.; data analysis, interpretation, manuscript editing and review: S.T. (Sungcom Techawongstien), P.T., T.T. and S.T. (Suchila Techawongstien). All authors have read and agreed to the published version of the manuscript.

**Funding:** This research received no external funding.

**Institutional Review Board Statement:** Not applicable.

**Informed Consent Statement:** Not applicable.

**Data Availability Statement:** Not applicable.

**Acknowledgments:** The authors are grateful to The Royal Golden Jubilee (RGJ) Ph.D. program, Thailand Science Research and Innovation (TSRI), and the Research Group for Fruit Crops in the Northeast, Khon Kaen University. We also thank the Plant Breeding Research Center for Sustainable Agriculture at Khon Kaen University, as well as the Laboratory of Pomology, Kyoto University for facilities support.

**Conflicts of Interest:** The authors declare no conflict of interest.

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
