# Peer review of "The Responses of Physiological Characteristics and Flowering Related Gene to the Different Water Stress Levels of Red-Flesh Pummelo Cultivars (Citrus grandis (L.) Osbeck) Own-Rooted by Air Layering Propagation under Two Growing Conditions"

_horticulturae, doi:10.3390/horticulturae7120579_

Round 1

Reviewer 1 Report

1- Abstract must have quantitative results 

2- In introduction, lines 56-68, you should enhance this paragraph with recent publications about water stress and induced negative physiological impacts on plants. For help, you may use the following references:

DOI: 10.1002/agj2.20846 ; https://doi.org/10.3390/plants9101346

3- Improve resolution of all figures

4- Discussion need recent references to compare with your findings in order to highlight your motivation 

4- Add key message on the end of conclusion

Author Response

Dear reviewer of Horticulturae,

            After getting your comments on our manuscript, we tried to revise all points followed the reviewers and your suggestions. After revision we found that the page number and lines have been changed, which we indicated in responses to the reviewers’ comments. 

Thank you very much for your kind considerations.

Sincerely Yours,

Suchila Techawongstien

Professor, Department of Plant Science and Agricultural Resources

Faculty of Agriculture, Khon Kaen University, Khon Kaen, 40002

Thailand

Responses to reviewers comments

Reviewer 1

            Thank you for your comments to make our manuscript more readable and better understood by the reader. We have corrected many of your suggestions.  Please find below our explanations for not changing some aspects of our manuscript.  

Question 1 : Abstract must have quantitative results.

Answer 1 : We already corrected as your suggestions.

            Page 1, line 27: We agree with the suggestion and have added “225 flowers”

            Page 1, line 28: We agree with the suggestion and have added “1.48%”

            Page 1, line 28: We agree with the suggestion and have added “50.53 m-2s-1

            Page 1, line 29: We agree with the suggestion and have added “0.30 Fv/Fm”

            Page 1, line 29: We agree with the suggestion and have added “13.95”

Question 2 : In introduction, lines 56-68, you should enhance this paragraph with recent publications about water stress and induced negative physiological impacts on plants. For help, you may use the following references:

https://doi.org/10.1002/agj2.20846 ; https://doi.org/10.3390/plants9101346

Answer 2 : We already corrected following your suggestions and have added.

            Page 2, line 52-72 : The information about water stress and induced negative physiological impacts on plants.

Question 3 : Improve resolution of all figures.

Answer 2 :  We agree with the suggestion and have added provides clear illustrations.

            Page 4, line 156 : figure 1 provides clear illustrations (900 pixels/cm)

            Page 5, line 175 : figure 2 provides clear illustrations (900 pixels/cm)

            Page 7, line 243 : figure 3 provides clear illustrations (900 pixels/cm)

Question 4 : Discussion need recent references to compare with your findings in order to highlight your motivation.

Answer 4 : We already cross checked all the references used in the text and have added recent references.

            Page 9, line 295: We agree with the suggestion and have added  https://doi.org/10.1093/jxb/erab048

            Page 9, line 296: We agree with the suggestion and have added https://doi.org/10.1007/s11248-021-00239-w

            Page 9, line 296: We agree with the suggestion and have added https://doi.org/10.1016/B978-0-12-817112-7.00013-4

            Page 9, line 302: We agree with the suggestion and have added https://doi.org/10.1016/j.plantsci.2021.111007

Question 5 : Add key message on the end of conclusion.

Answer 5 : We agree with the suggestion and have added the information for the key message of the conclusion in page 9 lines 313-320.

Reviewer 2 Report

Detailed comments are attached herewith.

Author Response

Dear reviewer of Horticulturae,

            After getting your comments on our manuscript, we tried to revise all points followed the reviewers and your suggestions. After revision we found that the page number and lines have been changed, which we indicated in responses to the reviewers’ comments. 

Thank you very much for your kind considerations.

Sincerely Yours,

Suchila Techawongstien

Professor, Department of Plant Science and Agricultural Resources

Faculty of Agriculture, Khon Kaen University, Khon Kaen, 40002

Thailand

Reviewer 2

Question 1 :  In the Introduction Section, only a few studies have been cited regarding physiological responses related to the flowering-related gene. It is suggested to add more relevant recent studies in the literature section. Also, the research gap should be clearly described

Answer 1 : We already corrected as your suggestions,

            Page 2, lines 51-72 : We agree with the suggestion and have added the information about  physiological responses related to the flowering-related gene.

Question 2 : It is suggested to mention the objectives of the study in point-wise form, instead of a paragraph form.

Answer 2 : After discussion with our co-authors we prefer to remain our objective due to the typical style of physiological study. However, we used “objective” instead of “aim” in page 2 lines 84-87

Question 3 : The manuscript includes some grammatical errors, which should be checked and corrected in the revised manuscript.

Answer 3 :Thank you for your suggestion, therefore this manuscript was already improved by native English speaker (Ms. Natalie Yunchalard (Master of Arts in English, KKU) and Mr. Peter Brian Setter.

Question 4 : Line 65-69: modify these lines to have more clarity regarding the term Water stress. Also, justify how the floral induction is related to water stress under different climatic conditions.

Answer 4 : We already modified following your suggestions.

            Page 2, line 52-72 : We added the information about floral induction related to water stress.

Question 5: Line 91-92: What is the field capacity value for the Pummelo plants during the experimental work?

Answer 5: In our experiment, optimal conditions for under container (plastic pots) condition; the field capacity of a mixed media is 29%. For field condition; the field capacity of a soil is 22% and the storage capacity of the soil is 20.43 cm.

Question 6: Line 107: describe the term “Stomatal conductance”??

Answer 6: Stomatal conductance measured by a porometer is the rate of CO2 entering, or water vapor exiting through stomata.  We have added information of stomatal conductance on page 3, line 114-1116, which referred from https://doi.org/10.1016/B978-0-12-394807-6.00087-3

Question 7:  The quality of the Figures should be improved, particularly Figure 1.

Answer 7: We agree with the suggestion and improved the better illustrations (900 pixels/cm) in figure 1 page 4,         line 157.

Question 8 : Line 301-305: Explain the possible reason, why the treatments in the field growing conditions for mild stress levels did not reach the critical threshold value.

Answer 8: In field-grown for mild stress levels did not reach the critical threshold value because 2 possible reason

                        1) Field-grown could not control the depth of root zone, thus pummelo can uptake water compared to the container-grown.

                        2) Mild stress levels was not severe for all physiological characteristics such as chlorophyll fluorescence stress gave the highest = 0.81 (Fv/Fm) but severe stress gave the lowest chlorophyll fluorescence = 0.55 (Fv/Fm).

Question 9 : The conclusion section could be much more informative and supported quantitatively.

Answer 9: We already added supported quantitatively information as your suggestions.

            Page 9, line 311: We agree with the suggestion and have added “1.48% ”

            Page 9, line 311: We agree with the suggestion and have added “50.53 m-2s-1

            Page 9, line 311: We agree with the suggestion and have added “0.30 Fv/Fm”

            Page 9, line 312: We agree with the suggestion and have added “167.32 mg/g DW”

            Page 9, line 312: We agree with the suggestion and have added “9.22% DW”

            Page 9, line 314: We agree with the suggestion and have added “13.95”

            Page 9, line 318: We agree with the suggestion and have added “225 flowers”

Reviewer 3 Report

Overall, this is a decent paper with useful and clear information, just some comments:

  1. This is a question more related to the whole study, rainfall and high humidity are the problems to cause flower drop, and in your study, water stress increased flower numbers, is there a balance to show which level of the water stress is better? And since your experiment only explore effects of water stress on flower, can you discuss any yield compromise due to water stress, especially with folded leave under water stress, plant photosynthesis will be significantly reduced, will this cause yield reduction?
  2. Line 101, did you keep the water stress constant from the beginning of the study to keep the leaf rolling index? What’s your detailed schedule for watering? For example, during the 2-month period, how many times you watered WS1, 2, 3, 4, 5 and what’s the interval and total amount of water input.
  3. Figure 3, better show significant letter in the figure.

Author Response

Dear reviewer of Horticulturae,

            After getting your comments on our manuscript, we tried to revise all points followed the reviewers and your suggestions. After revision we found that the page number and lines have been changed, which we indicated in responses to the reviewers’ comments. 

Thank you very much for your kind considerations.

Sincerely Yours,

Suchila Techawongstien

Professor, Department of Plant Science and Agricultural Resources

Faculty of Agriculture, Khon Kaen University, Khon Kaen, 40002

Thailand

Reviewer 3

Question 1.1 : This is a question more related to the whole study, rainfall and high humidity are the problems to cause flower drop, and in your study, water stress increased flower numbers, is there a balance to show which level of the water stress is better?

Answer 1.1: Regarding your question, Water stress induces both flowering and the up-regulation of the homolog of FT gene in sweet orange (Citrus sinensis) [13,19]. In our study, we found that WS4 and WS5 levels are good for producing high flowers and not reducing fruit yields. In addition, all of these information are existing in page 2 lines 59-67

Question 1.2: And since your experiment only explore effects of water stress on flower, can you discuss any yield compromise due to water stress, especially with folded leave under water stress, plant photosynthesis will be significantly reduced, will this cause yield reduction?

Answer 1.1: In this study, fruit yield (‘KKU-105’ at WS4 and WS5) were not reduced (data not shown) due to their leaves could recover within 1-2 weeks after re-watering, compared to the other treatments. Although, plant photosynthesis  (chlorophyll fluorescence) was reduced, then enhanced carbohydrate accumulation in the bud as well as promoted flowering [29] and not reducing fruit yields.

Question 2.1: Line 101, did you keep the water stress constant from the beginning of the study to keep the leaf rolling index? What’s your detailed schedule for watering?

Answer 2.1 : The information about  five levels of water stress was existing on page 4 line 157 and figure 1. Which leaf rolling index consist of five indices, i.e., unrolled leaves in the control group (WS1), folded deep-V-shaped leaves (WS2), fully-cupped U-shaped leaves (WS3), margin-touching O-shaped leaves (WS4), and tightly-rolled leaves (WS5).

Question 2.2: What’s your detailed schedule for watering? For example, during the 2-month period, how many times you watered WS1, 2, 3, 4, 5 and what’s the interval and total amount of water input.

Answer 2.2 :  In our schedule for watering in container and field conditions are different, i.e.,

            Container (plastic pots) condition; day of water deficit (related to leaf rolling index)

                        WS1 = watered daily, WS2 = 2 day, WS3 = 5 day, WS4 = 8 day, and WS5 = 14 day

            Field condition; day of water deficit (related to leaf rolling index)

                        WS1 = watered daily, WS2 = 5 day, WS3 = 12 day, WS4 = 21 day, and WS5 = 28 day

Question 3:  Figure 3, better show significant letter in the figure.

Answer 3: We already corrected as your suggestions.

            We agree with the suggestion and have added show significant letter in the figure 3  page 7, line 243.

Reviewer 4 Report

The current article entitled “The Responses of Physiological Characteristics and Flowering 2 Related Gene to the Different Water Stress Levels of Red-Flesh 3 Pummelo Cultivars (Citrus grandis (L.) Osbeck) under Two 4 Growing Conditions” By Thammatha et al., explains the role of genes related to flowering induction in pummelo under water stress. The whole manuscript needs extensive corrections and rewritten by a native English speaker. The findings of this study could be useful in understanding the response of genes to stress and its mechanism. However, some major issues need to be addressed before taking any decision.

Plagiarism:

Plagiarism of the current manuscript is 13%

  1. Introduction
    1. The introduction needs a correlation of different paragraphs and proper sequence. It should be written in a more clear and comprehensive way. There are so many problems with English grammar and sentence buildup.
    2. Some sentences are very long with confusing meanings. Please rewrite.
    3. Materials and methods:
  2. Plant materials and treatments: This section seems scary and unclear. I would suggest explaining your methodology in detail.
  3. How many plants per replication? Container conditions (Day/light hrs?  Humidity? Temperature?) and field growing conditions. Elaborate?
  4. Growth stage of plants at which stress applied? How old plants are used, whether in months or days?
  5. Line 87: “Under the container condition, the pummelo plants were transplanted….” Transplanted from where? At which stage?
  6. Lines 100-103: Levels of water stress. Please explain comprehensively. WS1-WS5
  7. Authors should give an experimental layout because the writing is very confusing. Writing needs severe improvement.

  1. Results
  2. there is a striking lack of connectors between sentences and leading to confusion.
  3. For all the results, authors should clearly write about the differences between control and stressed plants rather than using ambiguous terms.
  4. There is so much repetition of words in each sentence. Please rewrite
  5. Discussions
  6. Discussion can be divided into sub-sections rather than writing a long discussion.
  7. Overall results and discussion section need extensive revisions as there are so many confusions that are misleading.
  8. Improve discussion by avoiding repetition of results in this part.
  9. Discussion is very shallow and needs in-depth discussion with the recent literature published.
  10. In discussion, there is a lack of a mechanistic approach.

General comments:

  1. Abbreviate the words at the beginning and then use the abbreviations throughout the manuscript.
  2. Use homogenous terms for the explanation, don’t use multiple terms for the same purpose.
  3. Figures and tables should be next to the text where they are mentioned.
  4. Avoid formatting mistakes.

Author Response

Dear reviewer of Horticulturae,

            After getting your comments on our manuscript, we tried to revise all points followed the reviewers and your suggestions. After revision we found that the page number and lines have been changed, which we indicated in responses to the reviewers’ comments. 

Thank you very much for your kind considerations.

Sincerely Yours,

Suchila Techawongstien

Professor, Department of Plant Science and Agricultural Resources

Faculty of Agriculture, Khon Kaen University, Khon Kaen, 40002

Thailand

Reviewer 4

  1. Introduction

Question 1: Introduction needs correlation of different paragraphs and proper sequence. It should be written in a more clear and comprehensive way. There are so many problems with English grammar and sentence buildup.

Question 2: Line 52: “Whilst citrus trees are subjected to floral inductive” spelling mistake “while”

Question 3: Some sentences are very long with confusing meanings. Please rewrite.

Answer 1, 2 and 3: We already corrected following your suggestions.

            Page 2, line 34-86: written in a more clear and proper sequence.

            This manuscript was already improved by native English speaker (Ms. Natalie Yunchalard (Master of Arts in English, KKU) and Mr. Peter Brian Setter.

  1. Materials and methods:

 Question 1: Plant materials and treatments: This section seems scary and unclear. I would suggest explaining your methodology in detail.

Answer 1: We agree with your suggestions and rewritten as following information.

            - Two red-flesh pummelo cultivars, i.e., ‘KKU-105’ (5 years old ) and ‘Manee Esan’ (5 years old) were used. They were grown under two different conditions, i.e., container (plastic pots) conditions and field conditions.(page 2-3 lines 89-96)

            - Container (plastic pots) condition under net house, the pummelo plants were transplanted into plastic pots during the months of March-April 2014. Plastic pots size 100-L (0.53 m × 0.72 m; W × H).(page 2 lines 91-93)

          - Field conditions, the pummelo plants were transplanted into field with a spacing of 6x6 m during the months of March-April 2014.(page 2-3 lines 94-96)
-The study was carried out in a field and container (plastic pots) condition under net house at the experiment farm of Khon Kaen University, Thailand during the months of October to November 2018. (page 3 lines 101-103)

Question 2.1: How many plants per replication?

Answer 2.1: We added “three plants of cultivars were used in each replication” in page 3 lines 106-107.

Question 2.2: Container conditions (Day/light hrs? Humidity? Temperature?) and field growing conditions. Elaborate?

Answer 2.2: Since the air temperature, relative humidity, and rainfall  in plant-house and field-grown are almost similar, then information was shown in one figure (figure 2 ) and we added “almost similar, i.e.,”  in page 4 line 163.

Question 3: Growth stage of plants at which stress applied? How old plants are used, whether in
months or days?

Answer 3 : Growth stage of red-flesh pummelo is mature stages. The information about  pummelos used were 5 years old which its was existing on page 2 line 89 and figure 1.        

Question 4: Line 87: “Under the container condition, the pummelo plants were transplanted....” Transplanted                         from where? At which stage?
Answer 4 :  We added “transplanted into plastic pots  and transplanted into field during the months of March-April 2014” in page 2 line 91-92.

Question 5: Lines 100-103: Levels of water stress. Please explain comprehensively. WS1-WS5

Answer 5 : We already answer  as question 2.2 of reviewer 3 as following information, In our schedule for watering in container and field conditions are different, i.e.,

            Container (plastic pots) condition; day of water deficit (related to leaf rolling index)

                        WS1 = watered daily, WS2 = 2 day, WS3 = 5 day, WS4 = 8 day, and WS5 = 14 day

            Field condition; day of water deficit (related to leaf rolling index)

                        WS1 = watered daily, WS2 = 5 day, WS3 = 12 day, WS4 = 21 day, and WS5 = 28 day

The five levels of water stress based on leaf rolling index, i.e., unrolled leaves in the 101 control group (WS1), folded deep-V-shaped leaves (WS2), fully-cupped U-shaped leaves (WS3), margin-touching O-shaped leaves (WS4), and tightly-rolled leaves (WS5).

  1. Results

    Question 1: In result, there is a striking lack of connectors between sentences and leading to confusion.

    Answer 1 : We added some connectors between sentences, i.e.,                          

                        Page 4, line 166:  "On the other hand"

                         Page 4, line 171:  "Contrastingly"

                         Page 5, line 182:  "On the contrary"

                         Page 5, line 187:  “Furthermore"

                        Page 5, line 189:  "Similarly"

                         Page 5, line 191:  “However"

                        Page 5, line 199:  "In addition"

                        Page 6, line 208:  "In addition"

                        Page 5, line 213:  "However" 

                        Page 6, line 216: “Nevertheless"

Question 2: For all the results, authors should clearly write about the differences between control and stressed plants rather than using ambiguous terms.

Answer 2 : We agree your suggestions and added some information                                               

                                    Page 5, lines 190-192

                                    Page 6 , lines 207-210

                                    Page 6 , lines 215-218

Question 3: There is so much repetition of words in each sentence. Please rewrite

Answer 3: We already corrected as your suggestions.

                        Page 4, line 166:  "relative expression of CiFT gene “ changed as “CiFT mRNA level”

                         Page 4, line 169:  "under the container growing condition “ changed as “container-grown”

Round 2

Reviewer 4 Report

Authors have significantly improved the article and now it seems suitable for publication in the Horticulturae. 

Author Response

Dear reviewer of Horticulturae,

      Thank you very much for your kind considerations.